# Reconstituting the Mammalian Apoptotic Switch in Yeast

**DOI:** 10.3390/genes11020145

**Published:** 2020-01-29

**Authors:** Peter Polčic, Marek Mentel

**Affiliations:** Department of Biochemistry, Faculty of Natural Sciences, Comenius University in Bratislava, Mlynská dolina CH-1, Ilkovičova 6, 84215 Bratislava, Slovakia; marek.mentel@uniba.sk

**Keywords:** apoptosis, *Saccharomyces cerevisiae*, Bcl-2 protein family, Bax, Bcl-XL, BH3-only proteins, mitochondria

## Abstract

Proteins of the Bcl-2 family regulate the permeabilization of the mitochondrial outer membrane that represents a crucial irreversible step in the process of induction of apoptosis in mammalian cells. The family consists of both proapoptotic proteins that facilitate the membrane permeabilization and antiapoptotic proteins that prevent it in the absence of an apoptotic signal. The molecular mechanisms, by which these proteins interact with each other and with the mitochondrial membranes, however, remain under dispute. Although yeast do not have apparent homologues of these apoptotic regulators, yeast cells expressing mammalian members of the Bcl-2 family have proved to be a valuable model system, in which action of these proteins can be effectively studied. This review focuses on modeling the activity of proapoptotic as well as antiapoptotic proteins of the Bcl-2 family in yeast.

## 1. Introduction

Apoptosis is a prominent form of programmed cell death, in mammalian cells responsible for proper development, tissue maintenance and prevention of cell damage [1,2]. Precise regulation of apoptosis is necessary as both excessive and insufficient cell death can result in serious pathologies including numerous neurodegenerative or autoimmune disorders as well as cancer. Two main pathways control the apoptosis in mammalian cells. Extrinsic (receptor) pathway, by which immune cells induce apoptosis in damaged cells by ligation of receptors at the target cell surface, and intrinsic (mitochondrial) pathway, in which many diverse cell death-inducing signals are processed within the cell to converge to the permeabilization of mitochondrial membranes and the release of the cytochrome c from mitochondrial intermembrane space into the cytosol [3]. Once in the cytosol, cytochrome c binds to the apoptotic protease-activating factor 1 (Apaf-1) to induce its oligomerization into heptameric complex–apoptosome–that recruits and activates procaspase-9 to initiate the effector caspases that coordinate the apoptotic program [4]. The release of cytochrome c from mitochondria, which constitutes a commitment point of mitochondrial apoptotic pathway, is regulated by the action of proteins of the Bcl-2 family (Figure 1). 

The Bcl-2 family consists of proteins that can both promote or inhibit apoptosis [6]. Most of these proteins are constitutively present in the cell cytoplasm and react to many diverse cell survival or death-inducing signals, integrating them and translating them into the single universal apoptosis inducing event–the permeabilization of the outer mitochondrial membrane. Founding member of the family—Bcl-2 (B-cell lymphoma 2)—is an apoptosis-inhibiting protein encoded by a gene identified at the chromosomal breakpoint resulting from chromosomal translocation occurring in lymphocytes of most patients with follicular lymphoma [7,8,9]. In affected chromosomes, the proximity of immunoglobulin heavy chain gene promoter results in the overexpression of Bcl-2 granting the cell resistance to induction of cell death.

Members of the Bcl-2 family share a homology with Bcl-2 in at least one of four conserved motifs called BH (Bcl-2 homology) domains [10]. Presence of BH domains in individual Bcl-2 family member proteins correlates with their function and according to both, the presence of BH domains and the function, Bcl-2 family proteins are classified into three subfamilies (Figure 2) [11].

The proteins Bax (Bcl-2-associated protein X) [12], Bak (Bcl-2 homologous antagonist/killer) [13] and Bok (Bcl-2 ovarian killer) [14] are proapoptotic proteins that constitute a subfamily referred to as multidomain proapoptotic proteins as they contain three BH domains (BH1-BH3). The role of Bax and Bak in initiation of apoptosis are far more understood than that of Bok. Although their intracellular localization differs in the absence of cell death-inducing signal, inactive monomeric Bax is present in cytosol [15,16] and inactive Bak monomers are in outer mitochondrial membrane, upon activation both proteins oligomerize in outer mitochondrial membrane to directly participate on the membrane permeabilization, likely by itself forming the pore. The third multidomain, proapoptotic protein is far less characterized. A low level of Bok, a protein attached to membranes of the Golgi apparatus and endoplasmic reticulum (ER), is maintained in living cells by proteolysis. Events in ER and/or inhibition of proteasome may result in upregulation of Bok and its oligomerization and mitochondrial outer membrane permeabilization [17]. 

The subfamily of antiapoptotic proteins members inhibit the release of cytochrome c from mitochondria by preventing the mitochondrial membrane permeabilization in cells when no cell death-inducing signal is present. The subfamily consists of Bcl-2 protein itself and its closest homologues, e.g., Bcl-XL and Bcl-w, which contain all four types of BH domains (BH1-4). Presence of BH4 domain is required for their antiapoptotic activity and distinguishes these proteins from proapoptotic family members. Antiapoptotic Bcl-2 family proteins are located either in the outer mitochondrial membrane (e.g., Bcl-XL) or in all intracellular membranes (e.g., Bcl-2) [18].

Finally, the members of the third subfamily are proteins that only contain BH3 domain and are therefore collectively referred to as BH3-only proteins [19]. Individual members of this subfamily are either present in different cellular locations in an inactive form and are activated by specific cell-death inducing signals, mostly by posttranslational modifications, or are regulated at the level of transcription (e.g., Puma and Noxa) [20]. Activated BH3-only proteins are typically released from their cellular location and relocated to the surface of mitochondria where they interact with other (antiapoptotic and/or proapoptotic) Bcl-2 family members to induce the formation of a pore by Bax and Bak.

Three-dimensional structure of multiple Bcl-2 family proteins, including representatives of each subfamily, have been determined [21,22,23]. Interestingly, both pro- and antiapoptotic multidomain proteins share a common fold, containing two hydrophobic helices surrounded by six amphipathic helices and one C-terminal hydrophobic helix that may serve as a membrane anchor. A characteristic hydrophobic grove at the protein surface, formed by residues of BH1, BH2 and BH3 domains, is capable of binding of a BH3 domain of other family member protein, enabling the multidomain proteins to form dimers with any member of the family containing BH3 domain. BH3-only proteins, on the other hand, generally are intrinsically unstructured proteins, BH3 domains of which only form an alpha helix-upon interaction with binding partner (multidomain Bcl-2 family member) [24].

As there are multiple members of each subfamily expressed in mammalian cells and, depending on apoptotic signal, some of them are activated, the Bcl-2 family proteins by themselves represent a complicated network of interactions. Reducing the complexity of this system by techniques of gene disruption is not an easy task, more so because inactivation of key antiapoptotic family members would result in nonviable cells. Alternative in vivo model systems, thus, are desirable for elucidating the molecular mechanisms that underlie the activity of these proteins. Yeast *Saccharomyces cerevisiae* is a simple unicellular organism that has been employed as a traditional model of eukaryotic cell since the very beginning of molecular and cell biology research. With the powerful battery of genetic and biochemical methods it can be rivalled by no other eukaryotic system. This review focuses on employing yeast to study the interactions of proteins of the Bcl-2 family and mechanisms, by which these proteins permeabilize the mitochondrial membranes.

## 2. Bcl-2 Proteins Expressed in Yeast

Although *S. cerevisiae*, and likely also other yeast species, undergo regulated forms of cell death under specific conditions [25,26], the yeast genome does not contain genes encoding for homologues of mammalian Bcl-2 family proteins (for the comment on one putative exception see below). However, when interactions of Bcl-2 family members were studied in nineties using yeast two-hybrid system, the apparent toxicity of hybrid molecules containing multidomain proapoptotic protein Bax was observed [27]. This toxicity was suppressed when hybrids containing antiapoptotic protein (e.g., Bcl-XL) were coexpressed [27], indicating that the toxicity may reflect the proapoptotic activity of the Bax that can be inhibited by Bcl-XL as it is in mammalian cells. Indeed, the expression of native murine Bax was promptly shown to be toxic for yeast cells, with the toxicity of Bax depending on homodimerization and mitochondrial localization [28]. It was further shown that in yeast cells expressing Bax, the significant portion of cytochrome c is released from mitochondria into the cytosol [29], indicating that Bax retains its membrane permeabilizing activity when expressed in yeast. And finally, applying the patch clamping technique to membranes isolated from yeast cells expressing Bax revealed a presence of a high conductance channel with electrophysiological characteristics essentially identical to the one that was detected in cultured human fibroblasts, in which apoptosis was induced by the withdrawal of the growth factor [30]. The single-channel conductance of this channel corresponds to the pore diameter large enough to let the particles as large as cytochrome c to diffuse through the membrane [30]. 

In mammalian cells, the purpose of the permeabilization of mitochondrial membranes by Bax and Bak is to deliver a signal molecule—cytochrome c—to the cytosol where the apoptosome assembles from Apaf1-cytochrome c complexes and procaspase-9 [31]. As no Apaf1 or other components of mammalian apoptotic pathway downstream from cytochrome c release are present in yeast, it is interesting to know why yeast cells die when mitochondrial membranes get permeabilized. The cytochrome c itself was shown to be dispensable for Bax-mediated killing of yeast, as yeast strains with deletions of both apocytochrome c isogenes (*CYC1* and *CYC7*) are either equally sensitive to the expression of Bax as a wild type [32] or die slower [33], which may, however, reflect a different growth kinetics of the strains. This indicates that it may either be the membrane permeabilization itself or release of some other components from intermembrane space of mitochondria that interferes with cell viability. When import of proteins into mitochondria was followed in Bax-expressing yeast using GFP-tagged mitochondrial proteins, it was found that expression of Bax interferes with the import of mitochondrial carriers to the inner mitochondrial membrane while appears not to affect the import of proteins to the mitochondrial matrix [34]. The import of mitochondrial carriers strictly depends on the presence of several soluble small Tim proteins in the intermembrane space (e.g., Tim9, Tim10, Tim12). These proteins act as chaperones that deliver the mitochondrial carrier preproteins to the Tim22 complex that catalyzes their integration into the membrane, and the function of these proteins is essential both for mitochondrial biogenesis and cell survival [35]. It may thus be that it is the release of small Tim proteins from the intermembrane space leading to cease in the essential processes of mitochondrial biogenesis that result into cell death [34]. This is also consistent with the detection of small Tim proteins among the proteins that are released from Bax-permeabilized yeast mitochondria [36]. 

Analysis of Bax-induced cell dying in different mutant yeast strains also revealed that the kinetics of decrease in cell viability correlates with the growth rate, that is—the same number of generations is required for the given decrease of viability in cells growing at different rates [34]. This would suggest that as cells divide, some essential factor may get diluted to the level that does not support the further cell survival. If Bax interferes with mitochondrial biogenesis, functional mitochondria may represent this factor. It thus appears that Bax-induced dying of yeast cells results from the pore-forming activity of Bax leading to inability of cells to propagate functional mitochondria rather than from Bax inducing yeast cell death pathways.

Interestingly, the Bax-induced loss of cell viability in yeast may be accompanied with several hallmarks typical for mammalian apoptosis, including the exposition of phosphatidylserine and chromatin condensation [37]. It is, therefore, apparent that some of the yeast regulated cell death subroutines may be activated in dying cells. 

Taken together, experiments described above have shown that the yeast cells heterologously expressing Bcl-2 proteins represent a valid experimental system, in which Bcl-2 family proteins preserve their activity and act on conserved cellular components that likely directly correspond to their mammalian targets. Activity of these proteins can thus be simply followed as assessment of viability of yeast cells after inducing expression of these proteins, e.g., by assessing the ability of growth on the plates containing inducer of the expression of studied Bcl-2 family proteins.

If, indeed, this is the case, then yeast cells expressing mammalian Bcl-2 proteins also represent a relevant and convenient model system, in which interactions of Bcl-2 family proteins with their cellular targets can be studied. Following this logic, participation of many proteins considered to be involved in the process of permeabilization of mitochondrial membranes by Bcl-2 proteins have been investigated by analysing the effects of deletions of yeast genes encoding for yeast orthologues of candidate proteins on the phenotypes induced by the expression of Bax. Here it should be noted that even though this type of experiment follows a straightforward logic and may be technically easy to execute, one has to be cautious in interpretation as indirect effects of these deletions may often affect the phenotypes. Experiments employing this strategy, showing that mitochondrial components including voltage-dependent anion channel (VDAC), ATP/ADP carrier, cyclophilin D, components of mitochondrial protein-import machinery (TIM and TOM), cardiolipin and even cytochrome c are dispensable for killing the cells by the expression of Bax, have been reviewed elsewhere [38].

Interestingly, the recent experiments employing this approach have shown that contact sites of ER and mitochondria, known as MAM (mitochondria associated membranes), may be involved in translocation of Bax to the mitochondrial membrane [39]. The activity of Bax was investigated in the mutant with the deletion of *MDM34*, which encodes for the component of ERMES (ER-mitochondria encounter structure)—a protein complex that is required for the stability of MAM in yeast [40]. When the mutant forms of Bax that do not require activation (see below) were expressed in *∆mdm34* cells, their activity was substantially abolished as compared to the expression in wild type cells, suggesting that Bax may in fact enter the mitochondrial membranes *via* endoplasmic reticulum and MAM [39].

## 3. Bax and Bak Activation

In mammalian cells, Bax and Bak remain inactive until the presence of proapoptotic signal. Inactivity of Bax and Bak results both from the presence of antiapoptotic members of the Bcl-2 family as well as from inactivity of proapoptotic BH3-only proteins. Activation of BH3-only proteins in the reaction to the proapoptotic signal is required for the activation of Bax and Bak. 

Defining feature of the BH3-only subfamily is the presence of only one of the conserved BH motifs—BH3. In the absence of apoptotic signal, individual BH3-only proteins reside in different cellular locations and act as transducers of distinct cell death-inducing signals to Bax and Bak. In response to death signals, BH3-only proteins undergo activation, which usually involves a posttranslational modification, and translocate to mitochondrial surface where they induce the formation of pore by Bax and Bak [19,41]. Several models describing the activation of Bax and Bak by BH3-only proteins have been proposed. In all of these models BH3-only proteins modulate the activity of multidomain Bcl-2 family members by binding their BH3 domain to the hydrophobic groove on the surface of the binding partner (multidomain anti- or pro-apoptotic proteins). The ‘direct activation’ model presumes that BH3-only proteins bind to the inactive monomeric form of proapoptotic proteins Bax and Bak and activate them by inducing their oligomerization in the mitochondrial membrane, which leads to the formation of the pore and release of cytochrome c. On the other hand, in the ‘indirect model’, BH3-only proteins bind to antiapoptotic proteins and inhibit their antiapoptotic activity. As these two basic models are not mutually exclusive, models exist, in which some of BH3-only proteins, mostly referred to as ‘activators’, directly activate Bax and Bak while others, ‘sensitizers’, inhibit antiapoptotic proteins [11,41].

As it is described above, the expression of Bax in yeast in the absence of antiapoptotic proteins has been shown to result in mitochondrial membrane permeabilization and cell death. This would indicate that Bax is an intrinsically active protein and no activation is required for its activity when it is not inhibited by antiapoptotic proteins. A number of papers, however, report that when synthetic gene, corresponding to yeast codon usage-optimized version of human Bax, is expressed in yeast, inactive protein is produced and either coexpression with the BH3-only protein (e.g., Puma), or introduction of specific activating mutation into expressed Bax is required for its activity [42] (Figure 3). Though it has been suggested that the latter behaviour is typical for the native (non-tagged) human protein while constitutive activity is typically associated with tagged versions of Bax protein [42], no consensus on this issue exists as both modified (e.g., N-terminally HA-tagged) and unmodified (non-tagged) versions of human or murine proteins have been described to be constitutively active (e.g., [28,43,44,45,46,47]).

While one would assume that the setting, in which the activation of Bax by BH3-only proteins is required (for the sake of simplicity we will refer to this as to ‘activation-requiring’ Bax) reflects the situation observed in mammalian cells, recent experiments with knock-out mammalian cells simultaneously lacking all eight genes encoding for BH3-only proteins indicated that even in mammalian cells Bax may be activated by the inhibition of antiapoptotic proteins only when BH3-only proteins are absent [48,49]. Dispensability of BH3-only proteins for Bax activation, thus, appears to correlate with the behaviour of the ‘constitutively active’ version Bax in yeast. Yet it is still possible, that other proapoptotic Bcl-2 family members, such as proapoptotic splice variants of Bcl-XL (Bcl-XS), may substitute for the role of absent BH3-only proteins.

Nevertheless, using the yeast expression system, in which inactive Bax is produced by the expression of unmodified human Bax and further activation is required for the Bax activity (‘activation requiring’ Bax), make it possible to study the molecular nature of the Bax activation and the regions and specific amino acid residues within the Bax that participate on the activation. Several mutations that convert ‘activation requiring’ human Bax to mitochondrially localized active protein have been described (see [38] for review). These mutations affect either N-terminal sequence referred to as ART (for Apoptotic Regulation and Targeting) [50,51] or the mobility of C-terminal hydrophobic alpha helix [50,52,53] that is important for the insertion of Bax into mitochondrial membranes [21]. Recently it was found, moreover, that Bax-targeting into mitochondrial membranes may be affected (inhibited) by acetylation of N-terminus Bax [54]. Expression of Bax in yeast mutants with the deletion of gene encoding for corresponding acetylase (*NAA20*) produces unacetylated version of protein that is mitochondrially localized but is not active, indicating that mitochondrial targeting by itself may not be sufficient for activation of Bax in yeast [54]. Yet, this experiment indicates that ART (N-terminus) likely plays a key role in Bax activation.

Similarly to Bax, the activity of another multidomain proapoptotic protein–Bak–has been studied in yeast. The expression of Bak mostly induces phenotypes that are comparable to those induced by ‘constitutively active’ Bax (e.g., [44,55,56]). When untagged version of human Bak was expressed in yeast, it resulted in inactive protein [57]. However, unlike with Bax, there is no report describing the activation of such inactive protein (e.g., by BH3-only proteins). As there is not so many data reported for Bak as there is for Bax, one can yet hardly conclude whether activation of both proteins follows a common path or whether there are relevant differences in mechanistic details between activation of Bak and activation of Bax observable in yeast. Such differences would possibly reflect the different localization of proteins, since prior to activation in mammalian cells, Bak is localized in mitochondrial membrane while Bax is in the cytosol. Activation of Bax therefore likely includes steps that are not involved in Bak activation (e.g., membrane insertion). 

## 4. Inhibition of Bax and Bak by Antiapoptotic Proteins

In mammalian cells, antiapoptotic proteins of the Bcl-2 family inhibit the pore-forming activity of Bax and Bak when no cell death-inducing signals are present. This situation can be reconstructed in yeast when mammalian genes encoding for antiapoptotic proteins, e.g., Bcl-XL or Bcl-2, are co-expressed with the constitutively active version of Bax (or Bak). In this setting the expression of antiapoptotic protein can completely inhibit the activity of Bax (or Bak) as no release of cytochrome *c* is observed in these cells, and viability is generally the same as the viability of the host strain that does not express any of the Bcl-2 family proteins (e.g., [29,58]). As the nature of interaction between anti- a pro-apoptotic Bcl-2 family proteins that mediate the inhibition of pore formation are not fully understood, the simple model like this represents a valuable tool, in which interaction of these proteins can be studied.

Among the characteristic features of Bcl-2 family proteins is their ability to heterodimerize [59,60]. As already mentioned, in multidomain Bcl-2 family proteins, such as Bcl-2, Bcl-XL, Bax or Bak, a hydrophobic groove is formed on their surface by BH1, BH2 and BH3 domains [22]. Typically, this groove can bind BH3 domain of any member of the Bcl-2 family. The functional significance of heterodimerization in Bax/Bak-inhibiting activity of antiapoptotic proteins, however, have been questioned. The mutation introduced into BH3-domain of Bcl-XL that disrupt the ability of Bcl-XL to heterodimerize with Bax (substitution of tyrosine 101 by lysine) has no effect on ability of Bcl-XL to inhibit the proapoptotic activity of Bax, both in yeast and in cultured mammalian cells [61]. Another result that indicates that the Bax-inhibiting activity of Bcl-XL is heterodimerization-independent is a lack of a stoichiometric relation typical for dimerization. When different amounts of Bax and Bcl-XL were coexpressed in yeast, using the two independent inducible promoters and the same tag enabling the quantification of expressed proteins, the same amounts of Bcl-XL were found to be required for supporting the cell survival, independent of the amount of the expressed Bax. In these experiments, in extreme situations, the required amount of Bcl-XL was either higher than the amount of expressed Bax or the smaller amount of Bcl-XL could completely inhibit the high excess of Bax. The amount of Bcl-XL both required for and able to inhibit Bax remained unchanged when the mutant form of Bcl-XL, unable to heterodimerize with Bax, was expressed [58]. One of the mechanisms that would demonstrate this type of quantitative behaviour would involve a hypothetical target of these proteins in the mitochondrial membrane, physical binding of Bax to which would lead to the formation of a pore and to the membrane permeabilization. A high affinity binding of Bcl-XL to this target would prevent the association of target with Bax. In this model the required amount of Bxl-XL required for preventing the membrane permeabilization would reflect the amount of target in mitochondrial membranes rather than the amount of Bax [58].

In the recent model of interactions between Bax and antiapoptotic proteins, based on studies in mammalian cells and in vitro, the antiapoptotic proteins inhibit the insertion of Bax into mitochondrial membranes by retrotranslocating Bax from mitochondrial surface into the cytosol [62]. In this model, Bax is permanently targeted to mitochondria and retrotranslocated into cytosol by Bcl-XL. The predominant cytosolic localization of Bax in the absence of cell death-inducing signal or its mitochondrial localization in the presence of such a signal are achieved by changes in the rate of Bax retrotranslocation by Bcl-XL (e.g., by its inhibition by BH3-only proteins). A Bcl-XL-dependent decrease of mitochondrially localized Bax after inhibition of proteosynthesis that likely corresponds to retrotranslocation of Bax from mitochondria have also been observed in yeast cells [63], indicating that no other Bcl-2 family proteins or other mammalian apoptotic proteins not present in yeast are required for Bax retrotranslocation. If the retrotranslocation would be achieved by transient interaction of Bcl-XL with Bax, one would, however, expect that increasing the amount of expressed Bax would require increased amount of Bcl-XL required for the rescue. On the other hand, if another protein in the mitochondrial membrane would be required for the insertion of Bax and Bcl-XL would retrotranslocate Bax by disrupting their interaction, the amount of required Bcl-XL could again reflect the amount of this protein, which would be in the agreement with the situation observed in yeast [58].

A more complex model has been proposed, based on data obtained both with cultured mammalian cells and with yeast cells expressing native human—activation requiring—Bax [64]. While this form of Bax is only very weakly associated with mitochondria when expressed alone (in the absence of antiapoptotic proteins), the expression of Bcl-XL increases the fraction of Bax associated with mitochondria. Opposite effect was observed when the Bax activated by introduction of point mutation was expressed. In this case, the protein spontaneously localizes in mitochondria but coexpression of Bcl-XL reduces the fraction of Bax localized in mitochondria. In proposed model, Bcl-XL both displace Bax from mitochondria and at the same time translocate Bax into mitochondrial membrane priming it for the formation of the pore. These two antagonistic activities would generate dynamic system, in which Bax would cycle between soluble, membrane attached and primed form, which would likely be very sensitive to the activation by BH3-only proteins (Figure 3b). Consistent with this model, which requires two different types of interactions between Bax and Bcl-XL, is the finding that these two interactions can be genetically distinguished since the retrotranslocating activity was found to absent in the mutant form of Bcl-XL lacking twenty C-terminal amino acids, while this mutant retained it Bax priming activity [64].

Actually, the fact that the insertion of Bax into the membrane and the formation of a pore is not a simple one step event has also been corroborated by earlier observation in yeast. When the integration of Bax into the mitochondrial membranes was investigated in the presence of Bcl-XL, cells were shown to survive even in situation when a fraction of Bax integrated to the membranes in alkaline resistant fashion was higher than the total amount of Bax able to kill cells in the absence of Bcl-XL. This would suggest that at least two subpopulations of membrane-integrated Bax exist and that presence of Bcl-XL can prevent the transition of membrane-integrated Bax to the membrane-integrated pore-forming Bax [58]. It is possible, that membrane integrated fraction of Bax that does not form a pore corresponds to the transiently mitochondrial form of Bax that is subject of retrotranslocation Bcl-XL. Even though the observations described above are in apparent agreement and the events taking place in forming the pore by ‘constitutively active’ Bax likely overlap with those in the activation of ‘activation requiring’ Bax, one has to keep in mind that this particular step need not necessarily correspond and be careful in interpreting them as the same molecular event.

## 5. BH3-only Proteins

Generally, when BH3 only proteins are expressed in yeast in the absence of multidomain Bcl-2 family proteins, they do not induce any apparent phenotypes [65,66]. However, when they are coexpressed with the Bax or Bak and Bcl-XL or Bcl-2, they affect their activity. As individual BH3-only proteins often require the activation in mammalian cells, these requirements are indeed reflected in the yeast model. Therefore, in individual cases the expression of modified versions, corresponding to the active form of the protein, may be required to interact with coexpressed Bcl-2 family members and to produce corresponding phenotypes. For example, BH3-only protein Bid is in mammalian cells expressed as an inactive 21 kDa cytosolic protein. Activation of Bid is achieved by proteolytic cleavage by a caspase 8, which produces the active 15 kDa protein, referred to as tBid (truncated Bid) [67,68,69]. In order to analyse the activity of tBid in yeast, which are naturally devoid of caspases, the corresponding truncated version of Bid gene, therefore, has to be expressed.

Several reports describe the analyses of the function of BH3–only proteins in yeast. When BH3–only proteins tBid, Bim, Bmf, Bik and Noxa are coexpressed together with either Bax or Bak and Bcl-XL or Bcl-2, they induce cell death [65,66]. As this phenotype is strictly dependent on expression of Bax or Bak, the dying can be attributed to activation of either one of the two. In these experiments, when BH3-only proteins were coexpressed with Bax or Bak in the absence of antiapoptotic proteins, no contribution to the dying from BH3-proteins was observed. These data clearly indicate that tested BH3-only proteins are able to inhibit the antiapoptotic activity of Bcl-XL or Bcl-2 (Figure 3a). It also appears that they do not directly activate Bax and Bak. The latter is, however, hard to conclude because Bax and Bak expressed in this system are already active in the absence of BH3-only proteins when no antiapoptotic proteins are present.

In other reports, different results were obtained with some BH3-only proteins. Coexpressed Bad, Puma and Bim were found to potentiate the cell killing activity of active Bax [70,71,72] and Puma has also been shown to activate inactive ‘activation requiring’ Bax [73]. While there is no satisfactory explanation for different results in different settings employing constitutively active versions, it appears that the use of inactive Bax may have unmasked the direct interactions of BH3-only proteins with Bax, as these are not required when constitutively active Bax is used. Using of different versions of Bax may thus be a way to separate the direct effects of BH3-only proteins on Bax from indirect effects involving inhibition of antiapoptotic proteins.

As described above, using yeast expressing Bcl-2 family proteins to study the activity of BH3-only proteins appears to be useful for addressing some specific question concerning the activity of BH3-only proteins but did not reliably answer the question whether or not BH3-only proteins activate Bax and Bak by direct interaction. This answer may has come from the in vitro studies, in which pore-forming activity of purified Bax and BH3 peptides derived from BH3-only proteins have been investigated. In these experiments BH3 peptides have been shown to interact with Bax when their α-helical structure is stabilized by covalent crosslink. When these ‘stabilized α-helices of BCL-2 domains’ (SAHBs) are added to soluble Bax, they activate it by binding to a site other than the canonical BH3-binding groove [74]. In the actual model of Bax activation that reflects these data, activated BH3-only proteins first interact with Bax by binding to the noncanonical binding site, inducing a conformational change that enables Bax to oligomerize. The oligomerization of Bax is further under the check by antiapoptotic proteins, which are able to inhibit the oligomerization of Bax activated by the BH3-only protein. The inhibition of antiapoptotic proteins by BH3-only proteins is thus further required for the pore formation [75].

Here it should be noted that screening of budding yeast genome database has identified an open reading frame that encodes for the protein that contains a sequence resembling the BH3 domain. Named *YBH3*, for yeast BH3-only protein, it has been reported to be an endoplasmic reticulum (ER) membrane integral protein that is required for cell killing in several regulated yeast cell death scenarios, including cell death induced by expression of Bax [76]. On the other hand, other authors report that the same open reading frame is homologous to Bax inhibitor-1 (BI-1) and is supporting yeast survival in conditions of ER stress rather than cell death. They named the gene accordingly as *BXI1* (Bax-inhibitor 1) [77]. As the putative BH3-only domain in this protein is overlapping with transmembrane segment, is truncated because of its position at a C-terminus [11], and there are no proteins that would interact with BH3 domain known to be natively present in yeast, it is unlikely that this is an authentic BH3 domain.

## 6. Anticancer Drugs

As in the case of B-cell lymphoma, the high activity of antiapoptotic proteins, e.g., resulting from their excessive expression, is involved in maintaining survival of many tumors [78]. Inhibition of antiapoptotic activity of these proteins is thus a reasonable and promising strategy in cancer therapy, making the antiapoptotic proteins of Bcl-2 family a target of choice for anticancer drugs. Among these drugs are compounds that mimic the BH3 peptide and bind to the hydrophobic groove of these proteins (See [79] for review). Several of these compounds were shown to be effective in inducing apoptosis in various mammalian cancer models or are in clinical trials, while one of them—Venetoclax (ABT-199) was already approved for treatment of chronic lymphocytic leukaemia (CLL) and small lymphocytic lymphoma (SLL) [80].

As the survival of different cancer cells depends on different specific antiapoptotic proteins, an easy method for testing the effect of these compounds in vivo, in system with individual antiapoptotic Bcl-2 family members, appears to be very useful. The effect of several BH3-mimetic compounds was tested in yeast cells coexpressing Bax together with different individual antiapoptotic proteins of the Bcl-2 family including Bcl-XL, Bcl-2, Bcl-w, Mcl-1 [81]. Two of tested drugs (ABT-737 and ABT-263) were shown to inhibit cell growth both on solid and liquid media in Bcl-2 family protein-dependent manner. Differences in sensitivity of strains expressing different antiapoptotic proteins corresponded to differences in their affinity to these proteins measured by in vitro techniques. Some of the tested compounds were found either toxic to yeast (oblatoclax mesylate, TW-37) or had no effect on the survival (HA14-1). These drugs, however, also induced nonspecific apoptotic activity in mammalian cells or were reported to kill mammalian cells only in very high concentration, respectively [82,83]. 

These experiments show that yeast expressing desired combination of Bcl-2 family proteins may in fact represent an advantageous model system that can be used for screening of chemical compounds for their cell death inducing activity. In the case of these experiments, no increase in sensitivity was observed in yeast strains defective in ABC transporters, indicating that efflux of these drugs from the cells is not a significant factor [81]. This may, however, not be a general rule and using of PDR-deficient strains for this type of experiments may be considered.

## 7. Conclusions

Since the discovery of Bcl-2, thirty-five years ago, the action of proteins constituting the Bcl-2 protein family is intensively studied. Although tremendous amount of data has been collected, not all are consistent when judged from the perspective of our current knowledge and we still do not have a satisfactory model of action of these proteins. This is partly due to the complexity of the network of interacting proteins and partly to the methodological limitations in experimenting with mammalian models. The yeast expressing mammalian apoptotic regulators of the Bcl-2 family represents a potent and simple in vivo experimental system, in which mutual interactions of these proteins and their interactions with other cellular components can be investigated. Several such expression systems that have been described and are available, however, differ in their behaviour. Each of the two different modes of action of these proteins, reflect the situation also observed in mammalian cells. Even though there is no sufficient explanation for these differences yet, the individual expression systems have proved to be a valuable models, in which the specific aspects of these proteins activity can be separated and studied individually, e.g., interaction of BH3-only proteins with antiapoptotic proteins in the system, in which BH3-only proteins are coexpressed with constitutively active version of Bax and antiapoptotic protein, while their direct interaction with multidomain proapoptotic proteins in the system, in which only BH3-only protein and ‘activation requiring’ version of Bax is expressed. Identifying the factors that underlie the different behaviour of Bcl-2 family proteins in these settings in yeast would help us better interpret the data collected with yeast models and, moreover, would likely be also directly relevant to the complicated behaviour of these proteins in native mammalian cells.

## Figures and Tables

**Figure 1 genes-11-00145-f001:**
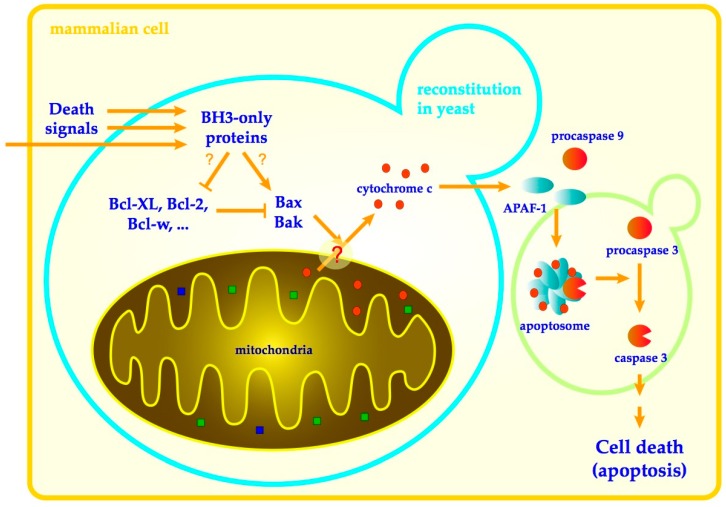
Mitochondrial (intrinsic) apoptotic pathway. Cell death-inducing signals are processed by proteins of the Bcl-2 family. BH3-only proteins in reaction to these signals activate Bax and Bak to permeabilize the outer mitochondrial membrane by either direct interaction or by inhibition of antiapoptotic proteins (e.g., Bcl-XL, Bcl-2). Cytochrome c escapes the permeabilized mitochondria to induce the assembly of apoptosome and activation of caspases, which ultimately results in cell death by apoptosis. The part of a pathway, reconstitution of which in yeast is a subject of this review, is outlined by light blue silhouette of budding yeast cell. Green silhouette of yeast cell outlines a part of the pathway that has been separately reconstituted in yeast [5] but is outside the scope of this review.

**Figure 2 genes-11-00145-f002:**
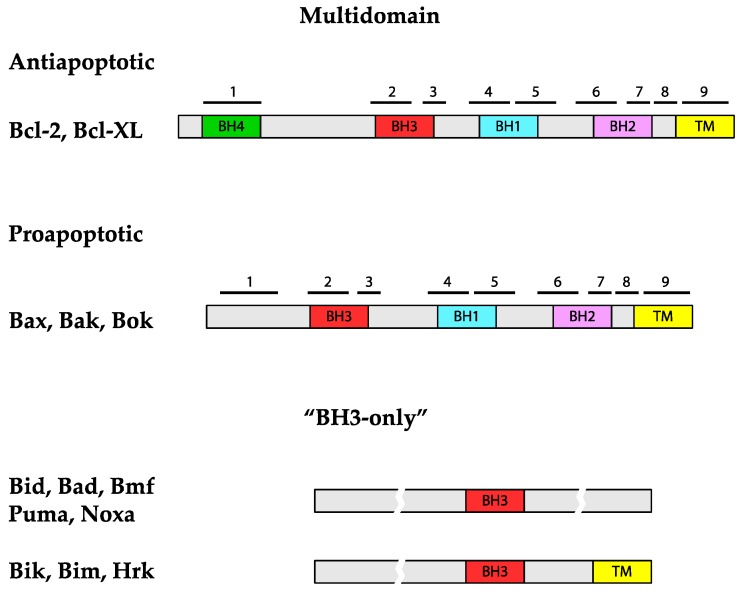
The Bcl-2 family of proteins. Domain structure of Bcl-2 family proteins is presented schematically. Sizes of proteins and domains are roughly in scale and position of BH domains is indicated. In multidomain proteins, positions of α-helices are indicated with numbered black bars.

**Figure 3 genes-11-00145-f003:**
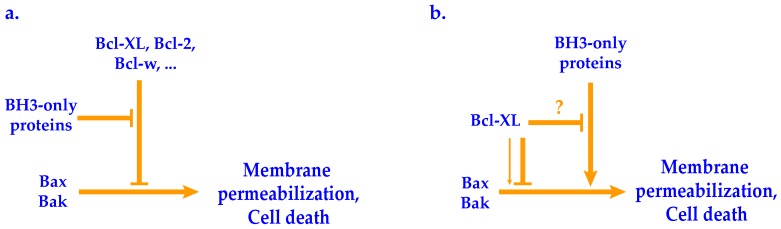
Two modes of Bcl-2 family proteins action observed in yeast model systems. (**a**) Bax or Bak are expressed as constitutively active cells killing proteins. Their activity is inhibited by coexpression of antiapoptotic protein (Bcl-XL, Bcl-2, Bcl-w). Coexpressed BH3-only proteins inhibit Bax-inhibiting activity of antiapoptotic proteins, resulting in active Bax or Bak. (e.g., [65]) (**b**) Bax requires activation by BH3-only proteins to become an active cells killing protein. (Such activation can also be achieved by introducing a point mutation into Bax). Bcl-XL appears to both promote mitochondrial targeting of Bax and inhibition of pore formation [63].

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
