# Peer review of "Reconstituting the Mammalian Apoptotic Switch in Yeast"

_genes, 2020, doi:10.3390/genes11020145_

Round 1

Reviewer 1 Report

The review adequately summarized the state of the field and highlighted questions that need to be addressed by those of us working in this area. It was balanced and comprehensive. A welcome review. 

Author Response

There are no specific points to be addressed raised by the reviewer.

Reviewer 2 Report

Polčic and Mentel’s review is comprehensive, well structured and clearly written. It introduces members of Bcl-2 family and describes how their activities are controlled in mammalian cells, and highlights aspects of their biological and biochemical functions that have been modeled in yeast. My only suggestions are to correct some typos, include some other references, and fix the bibliography so it matches the text.

Specific comments

Lines 33-36: Reference 4 is not an ideal citation for this statement about apoptosome formation and function. A suitable alternative may be PMID: 20844150.

Figure 1 is a nice illustration of the pathway and the aspects that are recapitulated in yeast. Although outside the scope of this review, readers may be interested to know that the Apaf-1/caspase-9/caspase-3 arm of the pathway has been separately reconstituted in yeast (see PMID: 11483856).

Lines 89-91. “Individual members of this subfamily are present in different cellular locations in an inactive form and are activated by specific cell-death inducing signals, mostly by posttranslational modifications.” While this is true for some BH-3-only proteins, the transcriptional regulation of Puma (and Noxa) by p53 is an important apoptotic trigger. I suggest this sentence is rephrased and the authors cite a review that describes the many processes by which BH3-only proteins are activated (eg PMID: 22492984).

Lines 254-255: “Unlike with Bax, there is no report describing the expression of protein that requires further activation.” This actually has been reported - PMID: 27108439 revealed that human Bak only killed yeast when N-terminally tagged.

Line 270 Typo: “As the nature of interaction between anti- a proapoptotic…” should presumably be “anti- and pro-apoptotic”

Lines 284-286 “the same amounts of Bcl-XL were found to be required for supporting the cell survival, independently form the amount of the expressed Bax.”. I suggest this is reworded to “…independent of the amount…”.

Line 304 Typo: “retrotralslocation” should be “retrotranslocation”

Line 317 Typo: “Opposite efect was observed” should be “…effect…”

Line 263: “5. Inhibition of Bax and Bak by antiapoptotic proteins”. This should be subsection 4, not 5.

The numbering throughout most of the bibliography doesn’t match the text – eg the final reference in the text is 78, which corresponds to the last paper in the bibliography, numbered as 75.

Author Response

Response to reviewer comments

We agree with all reviewer’s comments and we feel that following reviewer’s suggestions inproves the manuscript. We have made changes as described above:

Specific comments

Lines 33-36: Reference 4 is not an ideal citation for this statement about apoptosome formation and function. A suitable alternative may be PMID: 20844150.

We agree with the reviewer, we have replaced the citation as suugested. 

Figure 1 is a nice illustration of the pathway and the aspects that are recapitulated in yeast. Although outside the scope of this review, readers may be interested to know that the Apaf-1/caspase-9/caspase-3 arm of the pathway has been separately reconstituted in yeast (see PMID: 11483856).

We agree with the reviewer, we have modified the figure 1 as suggested and included a reference.

Lines 89-91. “Individual members of this subfamily are present in different cellular locations in an inactive form and are activated by specific cell-death inducing signals, mostly by posttranslational modifications.” While this is true for some BH-3-only proteins, the transcriptional regulation of Puma (and Noxa) by p53 is an important apoptotic trigger. I suggest this sentence is rephrased and the authors cite a review that describes the many processes by which BH3-only proteins are activated (eg PMID: 22492984).

We agree with the reviewer, we have modified the text accordingly as suggested and included a reference. Text now reads „Individual members of this subfamily are either present in different cellular locations in an inactive form and are activated by specific cell-death inducing signals, mostly by posttranslational modifications, or are regulated at the level of transcription (e.g. Puma and Noxa) [20].“

Lines 254-255: “Unlike with Bax, there is no report describing the expression of protein that requires further activation.” This actually has been reported - PMID: 27108439 revealed that human Bak only killed yeast when N-terminally tagged.

We have modified the text and included a reference. Text now reads „The expression of Bak mostly induces phenotypes that are comparable to those induced by ‘constitutively active’ Bax (e.g. [44,55,56]). When untagged version of human Bak was expressed in yeast, it resulted in inactive protein [57]. However, unlike with Bax, there is no report describing the activation of such inactive protein (e.g. by BH3-only proteins). As there is not so many data reported for Bak as there is for Bax, one can yet hardly conclude whether activation of both proteins follows a common path or whether there are relevant differences in mechanistic details between activation of Bak and activation of Bax observable in yeast.“.

Line 270 Typo: “As the nature of interaction between anti- a proapoptotic…” should presumably be “anti- and pro-apoptotic”

Typo has been corrected. 

Lines 284-286 “the same amounts of Bcl-XL were found to be required for supporting the cell survival, independently form the amount of the expressed Bax.”. I suggest this is reworded to “…independent of the amount…”.

Changed as suggested by reviewer.

Line 304 Typo: “retrotralslocation” should be “retrotranslocation”

Typo has been corrected. 

Line 317 Typo: “Opposite efect was observed” should be “…effect…”

Typo has been corrected. 

Line 263: “5. Inhibition of Bax and Bak by antiapoptotic proteins”. This should be subsection 4, not 5.

Typo in a subsection number has been corrected. 

The numbering throughout most of the bibliography doesn’t match the text – eg the final reference in the text is 78, which corresponds to the last paper in the bibliography, numbered as 75.

Error resulted from accidental inclusion of incorrect (old) version of the list of references in the final manuscript. We have corrected the error in revised version.

This manuscript is a resubmission of an earlier submission. The following is a list of the peer review reports and author responses from that submission.